# RE-PARAMETERIZING YOUR OPTIMIZERS RATHER THAN ARCHITECTURES

**Xiaohan Ding[1]\* Honghao Chen[2,3]\* Xiangyu Zhang[4,5]† Kaiqi Huang[2,3] Jungong Han[6] Guiguang Ding[7]‡**
[1]Tencent AI Lab      [2]CRISE, Institute of Automation, Chinese Academy of Sciences
[3]School of Artificial Intelligence, University of Chinese Academy of Sciences      [4]MEGVII Technology
[5]Beijing Academy of Artificial Intelligence    [6]Department of Computer Science, the University of Sheffield
[7]School of Software, BNRist, Tsinghua University
xiaohding@gmail.com    chenhonghao2021@ia.ac.cn    zhangxiangyu@megvii.com
kaiqi.huang@nlpr.ia.ac.cn    jungonghan77@gmail.com    dinggg@tsinghua.edu.cn

## ABSTRACT

The well-designed structures in neural networks reflect the prior knowledge incorporated into the models. However, though different models have various priors, we are used to training them with model-agnostic optimizers such as SGD. In this paper, we propose to incorporate model-specific prior knowledge into optimizers by modifying the gradients according to a set of model-specific hyper-parameters. Such a methodology is referred to as Gradient Re-parameterization, and the optimizers are named RepOptimizers. For the extreme simplicity of model structure, we focus on a VGG-style plain model and showcase that such a simple model trained with a RepOptimizer, which is referred to as RepOpt-VGG, performs on par with or better than the recent well-designed models. From a practical perspective, RepOpt-VGG is a favorable base model because of its simple structure, high inference speed and training efficiency. Compared to Structural Re-parameterization, which adds priors into models via constructing extra training-time structures, RepOptimizers require no extra forward/backward computations and solve the problem of quantization. We hope to spark further research beyond the realms of model structure design. Code and models https://github.com/DingXiaoH/RepOptimizers.

## 1 INTRODUCTION

The structural designs of neural networks are prior knowledge [1] incorporated into the model. For example, modeling the feature transformation as residual-addition ($y = x + f(x)$) outperforms the plain form ($y = f(x)$) (He et al., 2016), and ResNet incorporated such prior knowledge into models via shortcut structures. The recent advancements in structures have demonstrated the high-quality structural priors are vital to neural networks, e.g., EfficientNet (Tan & Le, 2019) obtained a set of structural hyper-parameters via architecture search and compound scaling, which served as the prior knowledge for constructing the model. Naturally, better structural priors result in higher performance.

Except for structural designs, the optimization methods are also important, which include **1)** the first order methods such as SGD (Robbins & Monro, 1951) and its variants (Kingma & Ba, 2014; Duchi et al., 2011; Loshchilov & Hutter, 2017) heavily used with ConvNet, Transformer (Dosovitskiy et al., 2020) and MLP (Tolstikhin et al., 2021; Ding et al., 2022), **2)** the high-order methods (Shanno, 1970; Hu et al., 2019; Pajarinen et al., 2019) which calculate or approximate the Hessian matrix (Dennis & Moré, 1977; Roosta-Khorasani & Mahoney, 2019), and **3)** the derivative-free methods (Rios & Sahinidis, 2013; Berahas et al., 2019) for cases that the derivatives may not exist (Sun et al., 2019).

We note that **1)** though the advanced optimizers improve the training process in different ways, they have no prior knowledge *specific to the model* being optimized; **2)** though we keep incorporating

---

\*Equal contributions. This work was partly done during their internships at MEGVII Technology.
†Project leader. ‡Corresponding author.
[1]Prior knowledge refers to all information about the problem and the training data Krupka & Tishby (2007). Since we have not encountered any data sample while designing the model, the structural designs can be regarded as some inductive biases Mitchell (1980), which reflect our prior knowledge.

our up-to-date understandings into the models by designing advanced structures, we train them with optimizers like SGD (Robbins & Monro, 1951) and AdamW (Loshchilov & Hutter, 2017), which are *model-agnostic*. To explore another approach, we make the following two contributions.

**1) A methodology of incorporating the prior knowledge into a model-specific optimizer.** We focus on non-convex models like deep neural networks, so we only consider first-order gradient-based optimizers such as SGD and AdamW. We propose to incorporate the prior knowledge via *modifying the gradients according to a set of model-specific hyper-parameters* before updating the parameters. We refer to this methodology as **Gradient Re-parameterization (GR)** and the optimizers as **RepOptimizers**. This methodology differs from the other methods that introduce some extra parameters (e.g., adaptive learning rate (Kingma & Ba, 2014; Loshchilov & Hutter, 2017)) into the training process in that we re-parameterize the training dynamics according to some hyper-parameters derived from the model structure, but not the statistics obtained during training (e.g., the moving averages recorded by Momentum SGD and AdamW).

**2) A favorable base model.** To demonstrate the effectiveness of incorporating the prior knowledge into the optimizer, we naturally use a model without careful structural designs. We choose a VGG-style plain architecture with only a stack of 3×3 conv layers. It is even simpler than the original VGGNet (Simonyan & Zisserman, 2014) (which has max-pooling layers), and has long been considered inferior to well-designed models like EfficientNets, since the latter have more abundant structural priors. Impressively, such a simple model trained with RepOptimizers, which is referred to as **RepOpt-VGG**, can perform on par with or better than the well-designed models (Table 3).

We highlight the novelty of our work through a comparison to RepVGG (Ding et al., 2021). We adopt RepVGG as a baseline because it also produces powerful VGG-style models but with a different methodology. Specifically, targeting a plain *inference-time* architecture, which is referred to as the *target structure*, RepVGG constructs extra training-time structures and converts them afterwards into the target structure for deployment. The differences are summarized as follows (Fig. 1). **1)** Similar to the regular models like ResNet, RepVGG also adds priors into models with well-designed structures and uses a generic optimizer, but RepOpt-VGG adds priors into the optimizer. **2)** Compared to a RepOpt-VGG, though the converted RepVGG has the same inference-time structure, the training-time RepVGG is much more complicated and consumes more time and memory to train. In other words, a RepOpt-VGG is a real plain model during training, but a RepVGG is not. **3)** We extend and deepen Structural Re-parameterization (Ding et al., 2021), which improves the performance of a model by changing the training dynamics via extra structures. We show that changing the training dynamics with an optimizer has a similar effect but is more efficient.

Of note is that we design the behavior of the RepOptimizer following RepVGG simply for a fair comparison and other designs may work as well or better; from a broader perspective, we present a VGG-style model and an SGD-based RepOptimizer as an example, but the idea may generalize to other optimization methods or models, e.g., RepGhostNet (Chen et al., 2022) (Appendix D).

From a practical standpoint, RepOpt-VGG is also a favorable base model, which features both *efficient inference* and *efficient training*. **1)** As an extremely simple architecture, it features low memory consumption, high degree of parallelism (one big operator is more efficient than several small operators with the same FLOPs (Ma et al., 2018)), and greatly benefits from the highly optimized 3×3 conv (e.g., Winograd Algorithm (Lavin & Gray, 2016)). Better still, as the model only comprises one type of operator, we may integrate many 3×3 conv units onto a customized chip for even higher efficiency (Ding et al., 2021). **2)** Efficient training is of vital importance to the application scenarios where the computing resources are limited or we desire a fast delivery or rapid iteration of models, e.g., we may need to re-train the models every several days with the data recently collected. Table 2 shows the training speed of RepOpt-VGG is around $1.8\times$ as RepVGG. Similar to the inference, such simple models may be trained more efficiently with customized high-throughput training chips than a complicated model trained with general-purpose devices like GPU. **3)** Besides the training efficiency, RepOptimizers overcome a major weakness of Structural Re-parameterization: the problem of quantization. The inference-time RepVGG is difficult to quantize via Post-Training Quantization (PTQ). With simple INT8 PTQ, the accuracy of RepVGG on ImageNet (Deng et al., 2009) reduces to 54.55%. We will show RepOpt-VGG is friendly to quantization and reveal the problem of quantizing RepVGG results from the structural transformation of the trained model. We naturally solve this problem with RepOptimizers as *RepOpt-VGG undergoes no structural transformations at all*.

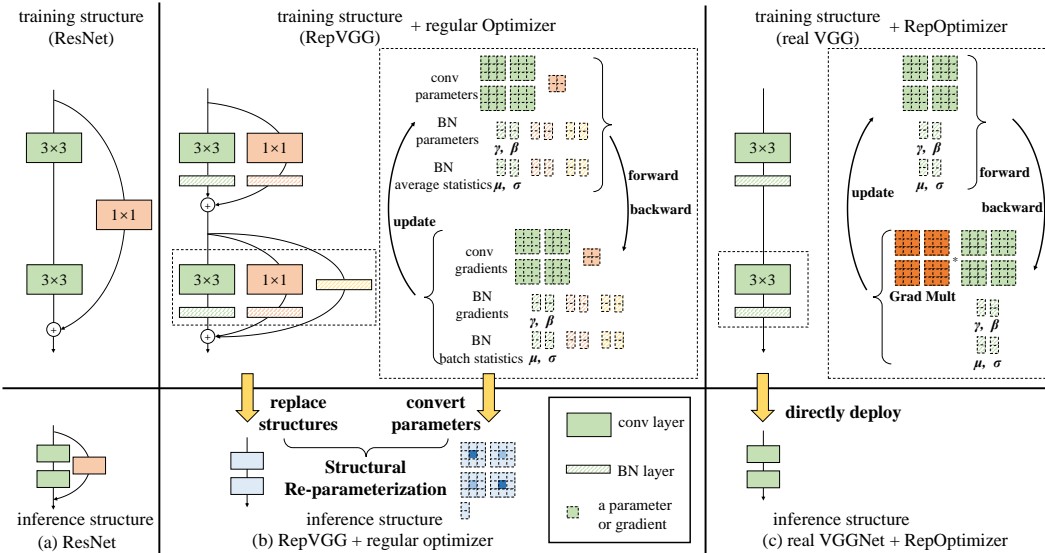

Figure 1: The differences among (a) a regular model, (b) a structurally-reparameterized RepVGG and (c) a real plain model trained with RepOptimizer. In this example, we assume the conv layers have two input/output channels so that a 3×3 layer has 2×2×3×3 parameters and a 1×1 layer has a parameter matrix of 2×2. The parameters of batch normalization (BN) (Ioffe & Szegedy, 2015) are denoted by $\gamma$ (scaling factor), $\beta$ (bias), $\mu$ (mean) and $\sigma$ (standard deviation). We use a rectangle to denote a trainable parameter as well as its gradient, or an accumulated BN parameter together with its batch-wise statistics. Please note that (a) has the same inference-time structure as its training-time structure and so does (c), but (b) does not. Though (b) and (c) have the same simple inference-time structure, all the extra structures and parameters of (b) requires forward/backward computations and memory during training, while (c) is simple even during training. The only computational overhead of (c) is an element-wise multiplication of the gradients by pre-computed multipliers *Grad Mult*.

## 2 RELATED WORK

**Structural Re-parameterization (SR).** RepVGG adopts *Structural Re-parameterization* (Ding et al., 2021), which converts structures via transforming the parameters. Targeting a VGG-style inference-time architecture, it constructs extra identity mappings and 1×1 conv layers for training and converts the whole block into a single 3×3 conv. In brief, the BN layers are fused into the preceding conv layers (note the identity mapping can also be viewed as a 1×1 conv layer whose kernel is an identity matrix), and the 1×1 conv kernels are added onto the central points of the 3×3 kernels (Fig. 1 b). A significant drawback of general SR is that the extra training costs cannot be spared. Compared to RepOpt-VGG, a RepVGG has the same target structure but complicated training structure.

**Classic re-parameterization.** GR extends and deepens SR just like SR extended the classic meaning of re-paramerization, e.g., DiracNet (Zagoruyko & Komodakis, 2017), which encodes a conv kernel in a different form for the ease of optimization. Specifically, a 3×3 conv kernel (viewed as a matrix) is re-parameterized as $\hat{W} = \text{diag}(\mathbf{a})I + \text{diag}(\mathbf{b})W_{\text{norm}}$, where $\mathbf{a}$, $\mathbf{b}$ are learned vectors, and $W_{\text{norm}}$ is the normalized trainable kernel. At each iteration, a conv layer first normalizes its kernel, compute $\hat{W}$ with $\mathbf{a}$, $\mathbf{b}$ and $W_{\text{norm}}$, then use the resultant kernel for convolution. Then the optimizer must derive the gradients of all the parameters and update them. In contrast, RepOptimizers do not change the form of trainable parameters and introduces no extra forward/backward computations.

**Other optimizer designs.** Recent advancements in optimizers include low-rank parameterization (Shazeer & Stern, 2018), better second-order approximation (Gupta et al., 2018), etc., which improve have no prior knowledge specific to the model structure. In contrast, we propose a novel perspective that some prior knowledge about architectural design may be translated into the design space of optimizer. Designing the behavior of optimizer is related to meta-learning methods that learn the update function (Vicol et al., 2021; Alber et al., 2018), which may be combined with our method because adding prior knowledge in our way may facilitate the learning process.

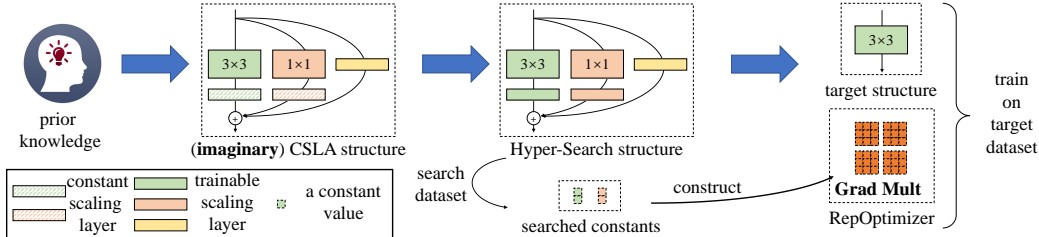

Figure 2: The pipeline of using RepOptimizers with RepOpt-VGG as an example.

# 3 REPOPTIMIZERS

RepOptimizers change the original training dynamics, but it seems difficult to imagine how the training dynamics should be changed to improve a given model. To design and describe the behavior of a RepOptimizer, we adopt a three-step pipeline: **1)** we define the prior knowledge and imagine a complicated structure to reflect the knowledge; **2)** we figure out how to implement *equivalent training dynamics* with a simpler target structure whose gradients are modified according to some hyper-parameters; **3)** we obtain the hyper-parameters to build up the RepOptimizer. Note the design of RepOptimizer depends on the given model and prior knowledge, and we showcase with RepOpt-VGG.

## 3.1 INCORPORATE KNOWLEDGE INTO STRUCTURE

The core of a RepOptimizer is the prior knowledge we desire to utilize. In the case of RepOpt-VGG, the knowledge is that a model's performance may be improved by *adding up the inputs and outputs of several branches weighted by diverse scales*. Such simple knowledge has sparked multiple structural designs. For examples, ResNet (He et al., 2016) can be viewed as a simple but successful application of such knowledge, which simply combines (adds up) the inputs and outputs of residual blocks; RepVGG (Ding et al., 2021) shares a similar idea but uses a different implementation. With such knowledge, we design structures to meet our demands. In this case, we desire to improve VGG-style models, so we choose the structural design of RepVGG ($3{\times}3$ conv + $1{\times}1$ conv + identity mapping).

## 3.2 SHIFT THE STRUCTURAL PRIORS INTO AN EQUIVALENT REPOPTIMIZER

Then we describe how to shift the structural prior into a RepOptimizer. We note an interesting phenomenon regarding the above-mentioned prior knowledge: in a special case where each branch only contains *one linear trainable operator* with optional *constant scales*, *the model's performance still improves* if the constant scale values are set properly. We refer to such a linear block as **Constant-Scale Linear Addition (CSLA)**. We note that we can replace a CSLA block by a *single operator* and realize *equivalent training dynamics* (i.e., they always produce identical outputs after any number of training iterations given the same training data) via multiplying the gradients by some constant multipliers derived from the constant scales. We refer to such multipliers as **Grad Mult** (Fig. 1). Modifying the gradients with Grad Mult can be viewed as a concrete implementation of GR. In brief,

$$\text{Proposition:} \qquad \text{CSLA block + regular optimizer = single operator + optimizer with GR.} \qquad (1)$$

We present the proof of such equivalency, which is referred to as *CSLA = GR*, in Appendix A. For the simplicity, here we only showcase the conclusion with two convolutions and two constant scalars as the scaling factors. Let $\alpha_A, \alpha_B$ be two constant scalars, $\mathrm{W}^{(A)}, \mathrm{W}^{(B)}$ be two conv kernels of the same shape, X and Y be the input and output, $*$ denote convolution, the computation flow of the CSLA block is formulated as $\mathrm{Y}_{CSLA} = \alpha_A(\mathrm{X} * \mathrm{W}^{(A)}) + \alpha_B(\mathrm{X} * \mathrm{W}^{(B)})$. For the GR counterpart, we directly train the target structure parameterized by $\mathrm{W}'$, so that $\mathrm{Y}_{GR} = \mathrm{X} * \mathrm{W}'$. Let $i$ be the number of training iterations, we can ensure $\mathrm{Y}_{CSLA}^{(i)} = \mathrm{Y}_{GR}^{(i)}, \forall i \geq 0$ as long as we follow two rules.

**Rule of Initialization**: $\mathrm{W}'$ should be initialized as $\mathrm{W}'^{(0)} \leftarrow \alpha_A \mathrm{W}^{(A)(0)} + \alpha_B \mathrm{W}^{(B)(0)}$. In other words, the GR counterpart should be initialized with the *equivalent parameters* (which can be easily obtained by the linearity) as the CSLA counterpart in order to make their initial outputs identical.

**Rule of Iteration**: while the CSLA counterpart is updated with regular SGD (optionally with momentum) update rule, the gradients of the GR counterpart should be multiplied by $(\alpha_A^2 + \alpha_B^2)$. Formally, let $L$ be the objective function, $\lambda$ be the learning rate, we should update $\mathrm{W}'$ by

$$W'^{(i+1)} \leftarrow W'^{(i)} - \lambda(\alpha_A^2 + \alpha_B^2)\frac{\partial L}{\partial W'^{(i)}} \,. \tag{2}$$

With *CSLA = GR*, we design and describe the behavior of RepOptimizer by first designing the CSLA structure. For RepOpt-VGG, the CSLA structure is instantiated by replacing the Batch Normalization (BN) following the $3 \times 3$ and $1 \times 1$ layers in a RepVGG block by a constant channel-wise scaling and the BN in the identity branch by a trainable channel-wise scaling (Fig. 2), since a CSLA branch has no more than one linear trainable operator. In such a slightly more complicated case with convolutions of different kernel sizes channel-wise constant scales, the Grad Mult is a tensor and the entries should be respectively computed with the scales on the corresponding positions. We present the formulation of the Grad Mult for training a single $3 \times 3$ conv corresponding to such a CSLA block. Let $C$ be the number of channels, $\mathbf{s}, \mathbf{t} \in \mathbb{R}^C$ be the constant channel-wise scales after the $3 \times 3$ and $1 \times 1$ layers, respectively, the Grad Mult $M^{C \times C \times 3 \times 3}$ is constructed by

$$M_{c,d,p,q} = \begin{cases} 1 + \mathbf{s}_c^2 + \mathbf{t}_c^2 & \text{if } c = d, p = 2 \text{ and } q = 2 \,, \\ \mathbf{s}_c^2 + \mathbf{t}_c^2 & \text{if } c \neq d, p = 2 \text{ and } q = 2 \,, \\ \mathbf{s}_c^2 & \text{elsewise} \,. \end{cases} \tag{3}$$

Intuitively, $p = 2$ and $q = 2$ means the central point of the $3 \times 3$ kernel is related to the $1 \times 1$ branch (just like a RepVGG block that merges the $1 \times 1$ conv into the central point of the $3 \times 3$ kernel). Since the trainable channel-wise scaling can be viewed as a "depthwise $1 \times 1$ conv" followed by a constant scaling factor $\mathbf{1}$, we add 1 to the Grad Mult at the "diagonal" positions (if the output shape does not match the input shape, the CSLA block will not have such a shortcut, so we simply ignore this term).

**Remark** Compared to the common SR forms like RepVGG, a CSLA block has no training-time nonlinearity like BN nor sequential trainable operators, and it can also be converted via common SR techniques into an equivalent structure that produce identical *inference* results. However, inference-time equivalency does not imply training-time equivalency, as the converted structure will have different training dynamics, breaking the equivalency after an update. Moreover, we must highlight that the CSLA structure is *imaginary*, which is merely an intermediate tool for describing and visualizing a RepOptimizer but we never actually train it, because the results would be mathematically identical to directly training the target structure with GR.

There remains only one question: how to obtain the constant scaling vectors $\mathbf{s}$ and $\mathbf{t}$?

### 3.3 OBTAIN THE HYPER-PARAMETERS OF REPOPTIMIZER VIA HYPER-SEARCH

Naturally, as the the hyper-parameters of RepOptimizer, Grad Mults affect the performance. Due to the non-convexity of optimization of deep neural network, the research community still lacks a thorough understanding of the black box, so we do not expect a provably optimal solution. We propose a novel method to associate the *hyper-parameters of an optimizer* with the *trainable parameters of an auxiliary model* and search, which is therefore referred to as **Hyper-Search (HS)**. Given a RepOptimizer, we construct the auxiliary Hyper-Search model by replacing the constants in *the RepOptimizer's corresponding CSLA model* with *trainable scales* and train it on a small search dataset (e.g., CIFAR-100 (Krizhevsky & Hinton, 2009)). Intuitively, HS is inspired by DARTS (Liu et al., 2018) that *the final values of trainable parameters are the values that the model expects them to become*, so the final values of the trainable scales are the expected constant values in the imaginary CSLA model. By *CSLA = GR*, the expected constants in the CSLA models are exactly what we need to construct the expected Grad Mults of the RepOptimizer. Note the high-level idea of HS is related to DARTS but we do not adopt the concrete designs of DARTS (e.g., optimizing the trainable and architectural parameters alternatively) since HS is merely an end-to-end training, not a NAS method.

### 3.4 TRAIN WITH REPOPTIMIZER

After Hyper-Search, we use the searched constants to construct the Grad Mults for each operator (each $3 \times 3$ layer in RepOpt-VGG) and store them in memory. During the training of the target model on the target dataset, the RepOptimizer element-wisely multiplies the Grad Mults onto the gradients of the corresponding operators after each regular forward/backward computation (*Rule of Iteration*).

We should also follow the *Rule of Initialization*. To start training with RepOptimizer, we re-initialize the model's parameters according to the searched hyper-parameters. In the case of RepOpt-VGG, just

Table 1: Architectural settings of RepOpt/RepVGG, including the number of 3×3 conv layers and channels of the four stages, and the theoretical inference-time FLOPs and number of parameters.

| Model | 3×3 layers | Channels | FLOPs (B) | Param (M) |
|---|---|---|---|---|
| B1 | 4, 6, 16, 1 | 128, 256, 512, 2048 | 11.9 | 51.8 |
| B2 | 4, 6, 16, 1 | 160, 320, 640, 2560 | 18.4 | 80.3 |
| L1 | 8, 14, 24, 1 | 128, 256, 512, 2048 | 21.0 | 76.0 |
| L2 | 8, 14, 24, 1 | 160, 320, 640, 2560 | 32.8 | 118.1 |

like the simplest example above ($\mathrm{W}'^{(0)} \leftarrow \alpha_A \mathrm{W}^{(A)(0)} + \alpha_B \mathrm{W}^{(B)(0)}$), we should also initialize each 3×3 kernel with the equivalent parameters of a corresponding CSLA block at initialization. More precisely, we suppose to initialize the 3×3 and 1×1 kernels in the imaginary CSLA block naturally by MSRA-initialization (He et al., 2015) as a common practice and the trainable channel-wise scaling layer as **1**. Let $\mathrm{W}^{(s)(0)} \in \mathbb{R}^{C \times C \times 3 \times 3}$ and $\mathrm{W}^{(t)(0)} \in \mathbb{R}^{C \times C \times 1 \times 1}$ be the randomly initialized 3×3 and 1×1 kernels, the equivalent kernel is given by (note the channel-wise scaling is a 1×1 conv initialized as an identity matrix, so we should add 1 to the diagonal central positions like Eq. 3)

$$\mathrm{W}'^{(0)}_{c,d,p,q} = \begin{cases} 1 + \mathbf{s}_c \mathrm{W}^{(s)(0)}_{c,d,p,q} + \mathbf{t}_c \mathrm{W}^{(t)(0)}_{c,d,1,1} & \text{if } c = d, p = 2 \text{ and } q = 2 \,, \\ \mathbf{s}_c \mathrm{W}^{(s)(0)}_{c,d,p,q} + \mathbf{t}_c \mathrm{W}^{(t)(0)}_{c,d,1,1} & \text{if } c \neq d, p = 2 \text{ and } q = 2 \,, \\ \mathbf{s}_c \mathrm{W}^{(s)(0)}_{c,d,p,q} & \text{elsewise} \,. \end{cases} \quad (4)$$

We use $\mathrm{W}'^{(0)}$ to initialize the 3×3 layer in RepOpt-VGG so the *training dynamics will be equivalent to a CSLA block initialized in the same way*. Without either of the two rules, the equivalency would break, so the performance would degrade (Table 4).

## 4 EXPERIMENTS

### 4.1 REPOPTIMERS FOR IMAGENET CLASSIFICATION

We adopt RepVGG as the baseline since it also realizes simple yet powerful ConvNets. We validate RepOptimizers by showing RepOpt-VGG closely matches the accuracy of RepVGG with much faster training speed and lower memory consumption, and performs on par with or better than EfficientNets, which are the state-of-the-art well-designed models. For a simpler example, we design RepOptimizer for RepGhostNet Chen et al. (2022) (a lightweight model with Structural Re-parameterization) to simplify the training model, eliminating structural transformations (Appendix D).

**Architectural setup.** For the fair comparison, RepOpt-VGG adopts the same simple architectural designs as RepVGG: apart from the first stride-2 3×3 conv, we divide multiple 3×3 layers into four stages, and the first layer of each stage has a stride of 2. Global average pooling and an FC layer follow the last stage. The number of layers and channels of each stage are shown in Table 1.

**Training setup.** For training RepOpt-VGG and RepVGG on ImageNet, we adopt the identical training settings. Specifically, we use 8 GPUs, a batch size of 32 per GPU, input resolution of 224×224, and a learning rate schedule with 5-epoch warm-up, initial value of 0.1 and cosine annealing for 120 epochs. For the data augmentation, we use a pipeline of random cropping, left-right flipping and RandAugment (Cubuk et al., 2020). We also use a label smoothing coefficient of 0.1. The regular SGD optimizers for the baseline models and the RepOptimizers for RepOpt-VGG use momentum of 0.9 and weight decay of $4 \times 10^{-5}$. We report the accuracy on the validation set.

**Hyper-Search (HS) setup.** We use CIFAR-100 for searching the hyper-parameters of RepOptimizers with the same configurations as ImageNet except 240 epochs and simple data augmentation (only left-right flipping and cropping). CIFAR-100 has only 50K images and an input resolution of 32, so the computational cost of HS compared to the ImageNet training is roughly only $\frac{50}{1281} \times \frac{240}{120} \times (\frac{32}{224})^2 \times \frac{3 \times 3 + 1 \times 1}{3 \times 3} = 0.18\%$. Note that the trainable scales of the HS model are initialized according to the layer's depth. Specifically, let $l$ be the layer's depth (i.e., the first layer in each stage has $l = 1$, the next layer has $l = 2$), the trainable scales are initialized as $\sqrt{\frac{2}{l}}$. The intuition behind this common trick is to let the deeper layers behave like an identity mapping at initialization to facilitate training (Goyal et al., 2017; Shao et al., 2020), which is discussed and analyzed in Appendix B.

**Comparisons with RepVGG.** First, we test the maximum batch size (MaxBS) per GPU to measure the training memory consumption and compare the training speed. That is, a batch size of MaxBS+1 would cause OOM (Out Of Memory) error on the 2080Ti GPU which has 11GB of memory. For the

Table 2: ImageNet accuracy and training speed (2080Ti, measured in samples/second/GPU).

| Model | Train MaxBS | Top-1 accuracy | | Train speed | | Params (M) | |
|---|---|---|---|---|---|---|---|
| | | @BS=32 | @MaxBS | @BS=32 | @MaxBS | train | inference |
| RepVGG-B1 | 198 | 78.41±0.01 | 78.65±0.03 | 213 | 243 | 57.4 | 51.8 |
| RepOpt-VGG-B1 | **260** | 78.47±0.03 | 78.62±0.03 | **380** | **445** | **51.8** | 51.8 |
| RepVGG-B2 | 153 | 79.53±0.08 | 79.78±0.14 | 142 | 163 | 89.0 | 80.3 |
| RepOpt-VGG-B2 | **200** | 79.36±0.09 | 79.66±0.12 | **246** | **264** | **80.3** | 80.3 |
| RepVGG-L1 | 106 | 79.39±0.07 | 79.84±0.02 | 124 | 136 | 84.3 | 76.0 |
| RepOpt-VGG-L1 | **140** | 79.46±0.02 | 79.83±0.03 | **221** | **252** | **76.0** | 76.0 |
| RepVGG-L2 | 77 | 80.52±0.05 | 80.56±0.09 | 82 | 86 | 131.0 | 118.1 |
| RepOpt-VGG-L2 | **103** | 80.29±0.02 | 80.53±0.09 | **138** | **149** | **118.1** | 118.1 |

Table 3: Comparisons among RepOpt-VGG, the original VGGNet-16 Simonyan & Zisserman (2014) and EfficientNets (Tan & Le, 2019). The inference speed is tested with FP32 on the same 2080Ti GPU. ‡ indicates extra SE Blocks (Hu et al., 2018). We also report the official EfficientNets trained with a much more consuming scheme (Tan & Le, 2019) in parenthesis.

| Model | Train epochs | Train resolution | Test resolution | Top-1 acc | Test batch size | Throughput (samples/second) |
|---|---|---|---|---|---|---|
| RepOpt-VGG-B1 | 120 | **224** | **224** | **78.47** | 128 | **970** |
| EfficientNet-B2 | 120 | 260 | 260 | 76.29 (79.8) | 128 | 912 |
| RepOpt-VGG-B2 | 120 | 224 | 224 | **79.39** | 128 | **645** |
| VGGNet-16 | 120 | 224 | 224 | 74.47 | 128 | 621 |
| RepOpt-VGG-L2 | 120 | **224** | **224** | **80.31** | 128 | **372** |
| EfficientNet-B3 | 120 | 300 | 300 | 79.12 (81.1) | 128 | 301 |
| RepOpt-VGG-L2‡ | **200** | **224** | 320 | **83.13** | 64 | 180 |
| EfficientNet-B4 | 300 | 380 | 380 | 82.02 (82.6) | 64 | 183 |
| EfficientNet-B5 | 300 | 456 | 456 | 82.96 (83.3) | 64 | 91 |

fair comparison, the training costs of all the models are tested with the same training script on the same machine with eight 2080Ti GPUs. The training speed of each model is tested with both the same batch size (32 per GPU) and the respective MaxBS. We train the models for 120 epochs both with a batch size of 32 and with their respective MaxBS. We linearly scale up the initial learning rate when enlarging the batch size. We repeat training each model three times and report the mean and standard deviation of accuacy. We have the following observations (Table 2).

**1)** RepOpt-VGG consumes less memory and trains faster: RepOpt-VGG-B1 trains $1.8\times$ as fast as RepVGG-B1 with the respective MaxBS. **2)** With larger batch size, the performance of every model raises by a clear margin, which may be explained by BN's better stability. This highlights the memory efficiency of RepOptimizers, which allows larger batch sizes. **3)** The accuracy of RepOpt-VGG closely matches RepVGG, showing a clearly better trade-off between training efficiency and accuracy.

**Comparisons with VGGNet-16 and EfficientNets.** As shown in Table 3, the superiority of RepOpt-VGG over the original VGGNet-16 trained with the identical settings can be explained by the depth (VGGNet-16 has 13 conv layers while RepOpt-VGG-B2 has 27) and better architectural design (e.g., replacing max-pooling with stride-2 conv). Then we compare RepOpt-VGG with EfficientNets, which have abundant high-quality structural priors. As EfficientNet (Tan & Le, 2019) did not report the number of epochs, we first train with the identical settings (BS=32) as described before. The input resolution is larger for EfficientNets because it is considered as an architectural hyper-parameter of EfficientNets. Table 3 shows RepOpt-VGG delivers a favorable speed-accuracy trade-off. For the EfficientNet-B4/B5 models, we extend the training epochs to 300 to narrow the gap between our implementation and the reported results (Tan & Le, 2019). Since EfficientNets use SE Blocks (Hu et al., 2018) and higher test resolution, we also increase the test resolution of RepOpt-VGG-L2 to 320 and insert SE Blocks after each $3\times3$ conv. Due to the limited GPU memory, we reduce the batch size while testing the throughput. To highlight the training efficiency of our model, we *still use training resolution of 224* and only train for 200 epochs. Impressively, RepOpt-VGG outperforms the official EfficientNet-B4, even though the latter consumed much more training resources.

## 4.2 ABLATION STUDIES

With RepOpt-VGG-B1, we first present baseline results with advanced optimizers (Shazeer & Stern, 2018; Loshchilov & Hutter, 2017; Kingma & Ba, 2014; Tieleman & Hinton, 2012), which can be viewed as re-parameterizing the training dynamics but without model-specific prior knowledge. Then

Table 4: Accuracy of RepOpt-VGG-B1 with different behaviors of optimizers.

| Optimizer | Source of constants | Change initialization | Modify gradients | Top-1 acc |
|---|---|---|---|---|
| SGD | N/A | | | 76.91 |
| RMSprop | N/A | | | 76.72 |
| Adam | N/A | | | 76.14 |
| AdamW | N/A | | | 75.03 |
| Adafactor | N/A | | | 74.69 |
| RepOptimizer | Hyper-Search | ✓ | ✓ | **78.47** |
| RepOptimizer | Hyper-Search | | ✓ | 77.42 |
| RepOptimizer | Hyper-Search | ✓ | | 77.07 |
| RepOptimizer | All **1** | ✓ | ✓ | 77.59 |
| RepOptimizer | Same as HS initialization | ✓ | ✓ | 77.71 |
| RepOptimizer | Average across channels | ✓ | ✓ | 77.40 |

Table 5: Transfer experiments with RepOpt-VGG-B1 on different search/target datasets.

| Target dataest | Search dataset | Top-1 acc on target dataset |
|---|---|---|
| | CIFAR-100 | 78.47 |
| ImageNet | ImageNet | 78.43 |
| | Caltech256 | 78.19 |
| | CIFAR-100 | 67.46 |
| Caltech256 | ImageNet | 67.55 |
| | Caltech256 | 67.42 |

we let the RepOptimizer not change the initialization or not modify the gradients. Table 4 shows either ablation degrades the performance because the training dynamics become no longer equivalent to training a CSLA model. Furthermore, we change the constant scales to **1)** all ones, **2)** same as the initial values ($\sqrt{\frac{2}{l}}$) in the HS model or **3)** the value obtained by averaging the scale values across all the channels for each layer. In all the cases, the performance degrades, suggesting that the diverse constant scales, which instruct the RepOptimizer to change the training dynamics differently for each channel, encode the vital model-specific knowledge. Note that our methodology generalizes beyond SGD, which applies to the other update rules like AdamW (please refer to the released code).

## 4.3 REPOPTIMIZERS ARE MODEL-SPECIFIC BUT DATASET-AGNOSTIC

The hyper-parameters searched on CIFAR-100 are transferrable to ImageNet, suggesting the RepOptimizers may be **model-specific** but **dataset-agnostic**. We further investigate by also Hyper-Searching on ImageNet and Caltech256 (Griffin et al., 2007) and using Caltech256 as another target dataset. The training and search settings on Caltech256 are the same as CIFAR-100, and the ImageNet search settings are the same as the ImageNet training settings. We have the following observations (Table 5). **1)** The hyper-parameters of RepOptimizers searched on the target datasets deliver no better results than those obtained on a different dataset: by searching and training both on ImageNet, the final accuracy is 78.43%, which closely matches the outcome of the CIFAR-searched RepOptimizer. Please note that searching on ImageNet does not mean training the same model twice on ImageNet because the second training only inherits the HS model's trained scales into the RepOptimizer but not the other trained parameters. **2)** With different hyper-parameter sources, RepOptimizers deliver similar results on the target datasets, suggesting that the RepOptimizers are dataset-agnostic.

Interestingly, the accuracy on the search dataset does not reflect the performance on the target dataset (Table 6). As RepOpt-VGG is designed for ImageNet, it has five stride-2 conv layers for downsampling. Training its corresponding HS model on CIFAR-100 seems unreasonable because CIFAR-100 has a resolution of 32×32, meaning the 3×3 conv layers in the last two stages operate on 2×2 and 1×1 feature maps. As expected, the accuracy of the HS model is low on CIFAR (54.53%), but the searched RepOptimizer works well on ImageNet. As we reduce the downsampling ratio of HS model on CIFAR-100 by re-configuring the five original downsampling layers to stride-1, the accuracy of HS model improves, but the corresponding RepOpt-VGG degrades. Such an observation serves as another evidence that *the searched constants are model-specific*, hence the RepOptimizers are model-specific: the original model has a total downsampling ratio of 32×, but setting a layer's stride to 1 makes a different HS model with 16× downsampling, so the constants searched by such an HS model predictably work poorly with the original 32×-downsampling model.

On downstream tasks including COCO detection (Lin et al., 2014) and Cityscapes (Cordts et al., 2016) semantic segmentation, RepOpt-VGG also performs on par with RepVGG (Table 7). For

Table 6: Results of changing the stride of the original five stride-2 layers suggesting the CIFAR-100 accuracy of the Hyper-Search model does not reflect the ImageNet accuracy of the target model.

| Stride | CIFAR-100 acc of Hyper-Search model | ImageNet acc of target model |
|---|---|---|
| 2, 2, 2, 2, 2 | 54.53 | 78.47 |
| 2, 2, 1, 2, 1 | 70.66 | 77.81 |
| 2, 1, 2, 1, 1 | 74.37 | 77.88 |

Table 7: Results of COCO detection and Cityscapes segmentation.

| Backbone | COCO mAP | Cityscapes mIoU |
|---|---|---|
| RepOpt-VGG-B1 | 43.50 | 79.36 |
| RepVGG-B1 | 43.60 | 79.15 |

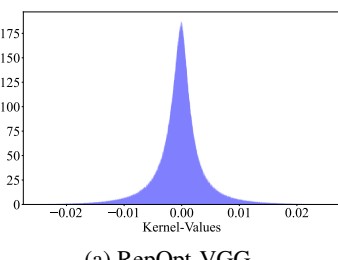 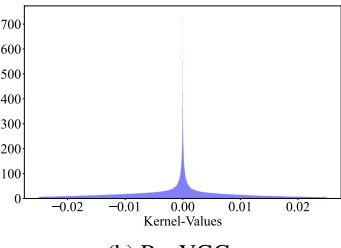

(a) RepOpt-VGG.                    (b) RepVGG.

Figure 3: Parameter distribution of the 3×3 kernels from RepOpt-VGG and inference-time RepVGG.

COCO, we use the implementation of Cascade Mask R-CNN (He et al., 2017; Cai & Vasconcelos, 2019) in MMDetection (Chen et al., 2019) and the 1x training schedule. For Cityscapes, we use UperNet (Xiao et al., 2018) and the 40K-iteration schedule in MMSegmentation (Contributors, 2020).

## 4.4 REPOPTIMIZERS FOR EASY QUANTIZATION

Quantizing a structurally re-parameterized model (i.e., a model converted from a training-time model with extra structures) may result in significant accuracy drop. For example, quantizing a RepVGG-B1 to INT8 with a simple PTQ (Post-Training Quantization) method reduces the top-1 accuracy to around 54.55%. Meanwhile, directly quantizing a RepOpt-VGG only results in 2.5% accuracy drop. We investigate the parameter distribution of the conv kernel of a RepOpt-VGG-B1 and an inference-time RepVGG-B1. Specifically, we first analyze the 8th 3×3 layer in the 16-layer stage of RepOpt-VGG-B1 and then its counterpart from RepVGG-B1. Fig. 3 and Table 8 shows the distributions of the kernel parameters from RepVGG-B1 are obviously different and the standard deviation is almost 4× as the RepOpt-VGG kernel. In Appendix C, we reveal that the structural conversion (i.e., BN fusion and branch addition) cause the quantization-unfriendly parameter distribution, and RepOptimizer naturally solves the problem because there is no structural conversion at all.

## 5 CONCLUSIONS AND LIMITATIONS

This paper proposes to shift a model's priors into an optimizer, which may spark further research beyond the realms of well-designed model structures. Besides empirical verification, we believe further investigations (e.g., a mathematically provable bound) will be useful, which may require a deeper understanding of the black box of deep neural networks. Another limitation is that our implementation presented in this paper relies on the *linearity* of operations as we desire to make the training dynamics of a single operator *equivalent* to a complicated block, but the methodology of shifting the structural priors into the training process may generalize beyond equivalently mimicking the training dynamics of another structure. For examples, we may use some information derived from the structure to guide the training, but do not have to ensure the equivalency to another model.

Table 8: Quantized accuracy and standard deviation of parameters from RepOpt-VGG or RepVGG.

| Model | Quantized accuracy | Standard deviation of parameters |
|---|---|---|
| RepOpt-VGG-B1 | 75.89 | 0.0066 |
| RepVGG-B1 | 54.55 | 0.0234 |

ACKNOWLEDGMENTS

This work is supported by the National Natural Science Foundation of China (Nos. 61925107, U1936202, 62021002), Beijing Natural Science Foundation (No. L223023), and the Beijing Academy of Artificial Intelligence (BAAI).

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

APPENDIX A: PROOF OF CSLA = GR

CSLA means each branch only comprises one differentiable linear operator with trainable parameters (e.g., conv, FC, scaling layer) and no training-time nonlinearity like BN or dropout. We note that *training a CSLA block with regular SGD is equivalent to training a single operator with modified gradients*.

We begin with a simple case where the CSLA block has two parallel conv kernels of the same shape each scaled by a constant scalar. Let $\alpha_A, \alpha_B$ be the two constant scalars, $W^{(B)}, W^{(B)}$ be the two conv kernels. Let X and Y be the input and output, the computation flow of the CSLA counterpart is formulated as $Y_{CSLA} = \alpha_A(X * W_A) + \alpha_B(X * W^{(B)})$, where $*$ denotes convolution. For the GR counterpart, we directly train the target structure parameterized by $W'$, so that $Y_{GR} = X * W'$. Assume the objective function is $L$, the number of training iterations is $i$, the gradient of a specific kernel W is $\frac{\partial L}{\partial W}$ and $F(\frac{\partial L}{\partial W'})$ denotes an arbitrary transformation on the gradient of the GR counterpart.

**Proposition**: there exists a transformation $F$ determined by only $\alpha_A$ and $\alpha_B$ so that updating $W'$ with $F(\frac{\partial L}{\partial W'})$ will ensure

$$Y_{CSLA}^{(i)} = Y_{GR}^{(i)} \quad \forall i \geq 0\,. \tag{5}$$

**Proof:** By the additivity and homogeneity of conv, we need to ensure

$$\alpha_A W^{(A)(i)} + \alpha_B W^{(B)(i)} = W'^{(i)} \quad \forall i \geq 0\,. \tag{6}$$

At iteration 0, a correct initialization ensures the equivalency: assume $W^{(A)(0)}$, $W^{(B)(0)}$ are the arbitrarily initialized values, the initial condition is

$$W'^{(0)} = \alpha_A W^{(A)(0)} + \alpha_B W^{(B)(0)}\,, \tag{7}$$

so that

$$Y_{CSLA}^{(0)} = Y_{GR}^{(0)}\,. \tag{8}$$

Then we use mathematical induction to show that with proper transformations on the gradients of $W'$, the equivalency always holds. Let $\lambda$ be the learning rate, the update rule is

$$W^{(i+1)} = W^{(i)} - \lambda \frac{\partial L}{\partial W^{(i)}} \quad \forall i \geq 0\,. \tag{9}$$

After updating the CSLA counterpart, we have

$$\alpha_A W^{(A)(i+1)} + \alpha_B W^{(B)(i+1)} = \alpha_A W^{(A)(i)} + \alpha_B W^{(B)(i)} - \lambda(\alpha_A \frac{\partial L}{\partial W^{(A)(i)}} + \alpha_B \frac{\partial L}{\partial W^{(B)(i)}}))\,. \tag{10}$$

We use $F(\frac{\partial L}{\partial W'})$ to update $W'$, which means

$$W'^{(i+1)} = W'^{(i)} - \lambda F(\frac{\partial L}{\partial W'^{(i)}})\,. \tag{11}$$

Assume the equivalency holds at iteration $i(i \geq 0)$. By Eq. 6, 10, 11, we must ensure

$$F(\frac{\partial L}{\partial W'^{(i)}}) = \alpha_A \frac{\partial L}{\partial W^{(A)(i)}} + \alpha_B \frac{\partial L}{\partial W^{(B)(i)}}\,. \tag{12}$$

Taking the partial derivatives on Eq. 6, we have

$$\frac{\partial W'^{(i)}}{\partial W^{(A)(i)}} = \alpha_A\,, \quad \frac{\partial W'^{(i)}}{\partial W^{(B)(i)}} = \alpha_B\,. \tag{13}$$

Then we arrive at

$$\begin{aligned} F(\frac{\partial L}{\partial W'^{(i)}}) &= \alpha_A \frac{\partial L}{\partial W'^{(i)}} \frac{\partial W'^{(i)}}{\partial W^{(A)(i)}} + \alpha_B \frac{\partial L}{\partial W'^{(i)}} \frac{\partial W'^{(i)}}{\partial W^{(B)(i)}} \\ &= (\alpha_A^2 + \alpha_B^2) \frac{\partial L}{\partial W'^{(i)}}\,, \end{aligned} \tag{14}$$

which is exactly the formula of the transformation.

With Eq. 14, we ensure $\alpha_A W^{(A)(i+1)} + \alpha_B W^{(B)(i+1)} = W'^{(i+1)}$, assuming $\alpha_A W^{(A)(i)} + \alpha_B W^{(B)(i)} = W'^{(i)}$. By the initial condition (Eq. 8) and the mathematical induction, the equivalency for any $i \geq 0$ is proved.

**Q.E.D.**

In conclusion, the gradients we use for updating the operator of the GR counterpart should be simply scaled by a constant factor, which is $(\alpha_A^2 + \alpha_B^2)$ in this case. By our definition, this is exactly the formulation we desire to construct the **Grad Mult** with the constant scales.

More precisely, with constants $\alpha_A$, $\alpha_B$ and two independently initialized (e.g., by MSRA-initialization He et al. (2015)) kernels $W^{(A)(0)}, W^{(B)(0)}$, the following two training processes always give mathematically identical outputs, given the same sequences of inputs.

**CSLA**: construct two conv layers respectively initialized as $W^{(A)(0)}$ and $W^{(B)(0)}$ and train the CSLA block ($Y_{CS} = \alpha_A(X * W_A) + \alpha_B(X * W_B)$).

**GR**: construct a single conv parameterized by $W'$, initialize $W'^{(0)} = W^{(A)(0)} + W^{(B)(0)}$, and modify the gradients by Eq. 14 before every update.

In this proof, we assume the constant scales are scalars and the two conv kernels are of the same shape for the simplicity, so the Grad Mult is a scalar. Otherwise, the Grad Mult will be a tensor and the entries should be respectively computed with the scales on the corresponding positions, just like the formulation to construct the Grad Mults for RepOpt-VGG, which is presented in the paper. Except for conv, this proof generalizes to any linear operator including FC and scaling.

## APPENDIX B: DISCUSSIONS OF THE INITIALIZATION

### INITIALIZATION OF THE HS MODEL

For Hyper-Search (HS), the trainable scales of the HS model are initialized according to the layer's depth. Let $l$ be the layer's depth (i.e., the first block in each stage that has an identity branch is indexed as $l = 1$, the next has $l = 2$), the trainable scales (denoted by $\mathbf{s}, \mathbf{t}$ in the paper) after the conv layers are initialized as $\sqrt{\frac{2}{l}}$. The trainable scales in the identity branches are all initialized as $\mathbf{1}$.

Such an initialization strategy is motivated by our observation: in a deep model with identity shortcuts (e.g., ResNet, RepVGG and an HS model of RepOpt-VGG), the shortcut's effect should be more significant at deeper layer at initialization. In other words, the deeper transformations should be more like an identity mapping to facilitate training Shao et al. (2020). Intuitively, in the early stage of training, the shallow layers learn most and the deep layers do not learn much Zeiler & Fergus (2014), because the deep layers cannot learn well without good features from shallow layers.

For a quantitative analysis, we construct a ResNet with a comparable architecture to RepOpt-VGG-B1, which means the ResNet's first three stages have 4, 6, 16 blocks, respectively. For the blocks within the 16-block stage (except the first one because it has no identity path), we record the variance of the identity path together with the variance of the sum, and compute the ratio of the former to the latter. Fig. 4 shows that the identity variance ratio is larger at deeper layers. Such a phenomenon is expected as the output of *every* residual block has a mean of 0 and a variance of 1 (because it goes through a batch normalization layer). After every shortcut-addition, the variance of the identity path increases, so that it becomes more and more significant compared to the parallel residual block. This discovery is consistent with a prior discovery Goyal et al. (2017) that initializing the scaling factor of the aforementioned batch normalization (BN) layer as 0 facilitates training.

We did not observe a similar pattern in the HS model. Predictably, as there is no BN before the addition, the variances of the three branches all become larger as the depth increases so that the identity variance ratio maintains. Therefore, in order to produce a similar pattern, we initialize the trainable scales after the conv layers in the HS model according to the depth. Specifically, by initializing the trainable scales as $\sqrt{\frac{2}{l}}$, we make the identity branch more significant at deeper layers (Fig. 4). By doing so, the HS model and the RepOpt-VGG model are both improved (Table 9).

Table 9: The accuracy of the HS model with different initialization of the trainable scales after conv layers and the corresponding RepOpt-VGG-B1 on ImageNet.

| Initialization | CIFAR-100 acc | ImageNet acc |
|---|---|---|
| All $\mathbf{1}$ | 47.70 | 77.59 |
| $\sqrt{\frac{2}{l}}$ | 54.53 | 78.47 |

Figure 4: The identity variance ratio (the ratio of the variance of the identity path to the variance of the sum). By initializing the trainable scales according to the depth, we produce a pattern similar to ResNet.

Of note is that such an initialization scheme may not be optimal and we do not seek for a better initialization since the performance on the target dataset is not closely related to the accuracy of the HS model, as shown in Sec. 4.3.

INITIALIZATION OF THE TARGET MODEL

Of note is that the target model on the target dataset is initialized independent of the initial values of the conv kernels in the HS model. In other words, the only knowledge inherited from the HS model is the trained scales and we do not need to record any other initial information of the HS model. More precisely, let the initial state (including the values of every randomly initialized conv kernel) of the HS model and the target model be $\hat{\Theta}$ and $\Theta'$, respectively, we note that though the RepOptimizer's hyper-parameters are obtained via a training process beginning with $\hat{\Theta}$ on the search dataset, it works well with $\Theta'$ on the target dataset. As shown in the paper, we simply initialize a 3×3 kernel in the RepOpt-VGG as the equivalent kernel of the imaginary CSLA block computed with the trained scales inherited from the HS model and two independently initialized conv kernels. This discovery, again, shows that the RepOptimizers are specific to model *structures*, not datasets *nor a certain initial state*.

APPENDIX C: DISCUSSIONS OF THE QUANTIZATION

QUANTIZATION RESULTS

Quantizing a structurally re-parameterized network may result in a significant drop in accuracy, which limits its deployment. For example, directly quantizing a RepVGG-B1 to INT8 with a simple Post-Training Quantization (PTQ) method reduces the top-1 accuracy to around 54.55%, making the model completely unusable. Even worse, as a re-parameterized RepVGG has no BN layers anymore (and it is tricky to add BN layers into it again without harming the performance), finetuning with a Quantization-Aware Training (QAT) method is not straightforward to implement. Currently,

Table 10: Quantization results on RepOpt-VGG and RepVGG via both Post-Training Quantization (PTQ) and Quantization-Aware Training (QAT).

| Model | PTQ | QAT |
|---|---|---|
| RepOpt-VGG-B1 | 75.89 | 78.24 |
| RepVGG-1 | 54.55 | N/A |

Table 11: The standard deviation of parameters from RepVGG (both before and after the structural transformations) and RepOpt-VGG.

| Model | Kernel | Overall | Central | Surrounding |
|---|---|---|---|---|
| | 3×3 kernel before fusing BN | 0.0131 | 0.0128 | 0.0131 |
| RepVGG-B1 | after fusing BN | 0.0176 | 0.0175 | 0.0177 |
| | after branch addition | 0.0236 | 0.0500 | 0.0177 |
| RepOpt-VGG-B1 | 3×3 kernel | 0.0066 | 0.0151 | 0.0045 |

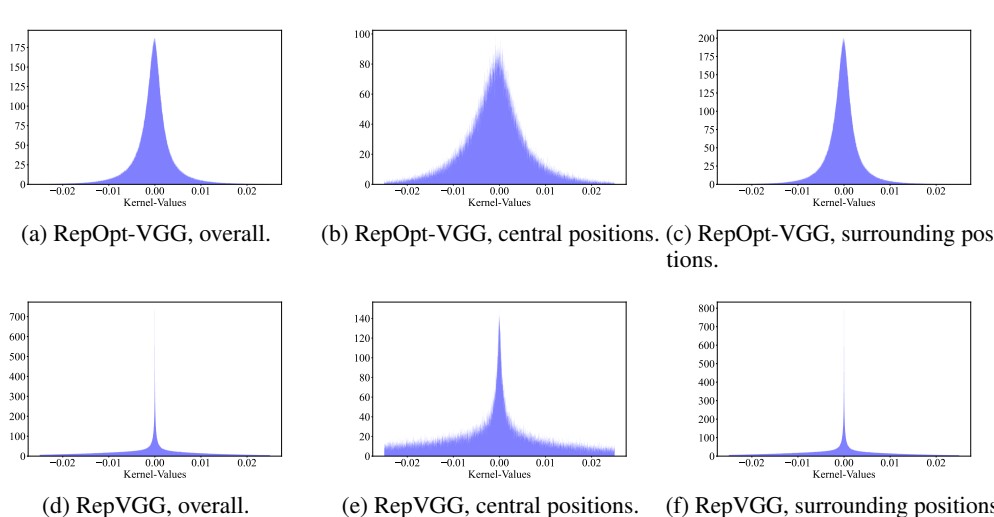

(a) RepOpt-VGG, overall.    (b) RepOpt-VGG, central positions. (c) RepOpt-VGG, surrounding positions.

(d) RepVGG, overall.    (e) RepVGG, central positions.    (f) RepVGG, surrounding positions.

Figure 5: Parameter distribution of the 3×3 kernel from RepVGG-B1 and RepOpt-VGG-B1 at different locations.

the quantization of structurally re-parameterized models remains an open problem, which may be addressed by a custom quantization strategy.

Compared to RepVGG, a RepOpt-VGG has the same inference-time structure. In contrast, we naturally solve the problem of quantization by completely eliminating the need for any structural transformations. For examples, a simple PTQ method provided by the PyTorch official library (torch.quantization) can quantize RepOpt-VGG-B1 to INT8 with only 2.58% drop in the accuracy (78.47% → 75.89%). And with a simple QAT method [1], we narrow the gap to only 0.23% (78.47%→78.24%) after only 10-epoch finetuning.

INVESTIGATION INTO PARAMETER DISTRIBUTION

To understand why RepVGG is difficult to quantize, we first investigate the parameter distribution of conv kernels of a RepOpt-VGG-B1 and an inference-time RepVGG-B1. Specifically, we sample the 8th 3×3 layer in the 16-layer stage of RepOpt-VGG-B1 together with its counterpart from RepVGG-B1. Fig. 3 and Table 8 in the paper show that the distribution of the kernel parameters of RepVGG-B1 is obviously different from the RepOpt-VGG kernel.

---

[1]Following the official example at https://pytorch.org/blog/introduction-to-quantization-on-pytorch/

Table 12: ImageNet accuracy of RepGhostNet and RepOpt-GhostNet.

| Model | Optimizer | Extra parallel fusion layers | Top-1 accuracy |
|---|---|---|---|
| RepGhostNet 0.5× | SGD | ✓ | 66.49 |
| RepOpt-GhostNet 0.5× | RepOptimizer | | 66.51 |

We continue to investigate the sources of such a high-variance parameter distribution of RepVGG. Since a training-time RepVGG block undergoes two transformations to finally become a single 3×3 conv, i.e., **1)** fusing BN and **2)** adding up branches, we evaluate the effects of the two transformations separately. Moreover, since the branch addition involves the identity branch, 1×1 conv and only the central points of the 3×3 kernels, we show the standard deviation of the central and the surrounding points (i.e., the eight non-central points in a 3×3 matrix) separately (Fig. 5, Table 11).

We make the following observations. **1)** In the 3×3 kernel sampled from RepVGG, the weights at different locations have similar standard deviation, and fusing BN enlarges the standard deviation. After branch addition, the standard deviation of the central points increases significantly, which is expected. **2)** The central points of the kernel form RepOpt-VGG have a larger standard deviation than the surrounding points (but lower than the converted kernel from RepVGG), which is expected because we use a Grad Mult with larger values at the central positions. **3)** The RepOpt-VGG kernel has much lower standard deviation than the converted RepVGG kernel, which is expected since it undergoes no BN fusion nor branch addition. **4)** Interestingly, even before BN fusion, the surrounding points of the 3×3 kernel in RepVGG has a larger variance than the counterparts in the RepOpt-VGG kernel. This can be explained that placing a BN after a conv can be viewed as multiplying a scaling factor to the conv's kernel and *the scaling factor varies significantly from batch to batch* (because the batch-wise mean and variance may differ significantly), leading to high variance of the conv kernel.

In summary, compared to Structural Re-parameterization, a significant strength of Gradient Re-parameterization is the capability to train a model without transforming the structures, which is not only efficient but also beneficial to the quantization performance.

## APPENDIX D: GENERALIZING TO REPGHOSTNET

As another example showing the potential generality of the proposed method, we use RepOptimizer to train a lightweight model named RepGhostNet Chen et al. (2022) without the original parallel fusion layers (implemented by Batch Normalization layers). More precisely, a RepGhostNet contains multiple RepGhost Modules, where we use fusion layers parallel to 1×1 convolutional layers (which are followed by Batch Normalization) to enrich the feature space. While designing the corresponding RepOptimizer, the CSLA structure is constructed by simply replacing the fusion layer with a trainable scaling layer and the Batch Normalization following 1×1 convolution by a constant scaling layer. Similar to RepOpt-VGG, we add a Batch Normalization layer after the addition of the two branches.

We experiment with RepGhostNet 0.5×. We search the scales of the two imaginary scaling layers for each target operator (i.e., 1×1 conv). Then we construct the Grad Mults, which can be viewed as a degraded case of RepOpt-VGG (from 3×3 to 1×1 and from three branches to two branches).

For training the target model on ImageNet, we follow Chen et al. (2022), extending the training schedule to 300 epochs, enlarging the learning rates and using simpler data augmentation. The RepGhostNet and corresponding RepOpt-GhostNet are trained with identical configurations. With RepOptimizer, We obtain an accuracy of 66.51, which closely matches that of our reproduced RepGhostNet (66.49), just like RepOpt-VGG closely matches RepVGG.

We have released the reproducible code and training scripts at `https://github.com/DingXiaoH/RepOptimizers`.

## APPENDIX E: RUN-TIME ANALYSIS OF SEARCHED SCALES

1) We depict the mean value of the scales obtained on different Hyper-Search datasets after each epoch. Specifically, we sample the scales from the last block of the third stage, including the scales

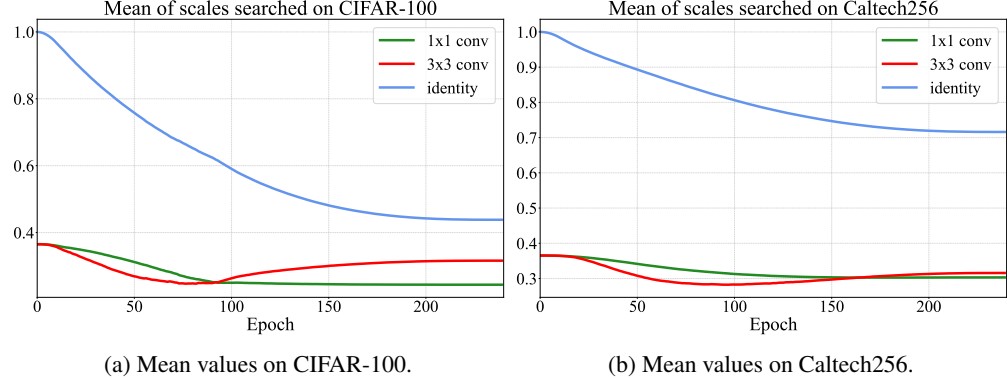

(a) Mean values on CIFAR-100.                    (b) Mean values on Caltech256.

Figure 6: Mean values of scales from the last block of the third stage during Hyper-Search on different datasets after each epoch. The scales searched on CIFAR-100 and Caltech256 show similar trends.

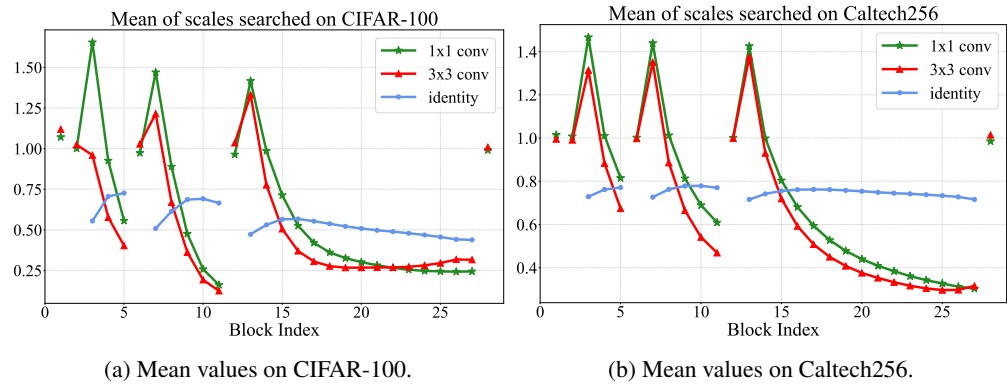

(a) Mean values on CIFAR-100.                    (b) Mean values on Caltech256.

Figure 7: Mean values of the trained scales from different blocks after Hyper-Search. The 1st, 2nd, 6th, 12th and last blocks have no identity branches (because they are the output does not match the shape of input) so that the curves are split, and a continuous curve represents a stage with more than one block. The scales searched on CIFAR-100 and Caltech256 show similar trends.

of the 1×1, 3×3 and identity branches. As shown in Fig. 6, the scales searched on two significantly different datasets show similar trends, which further support that the RepOptimizer is model-specific but dataset-agnostic.

2) In order to show how the searched scales vary with the depth of layers, we show the mean value of the searched scales after the last epoch on different Hyper-Search datasets. As shown in Fig. 7, the hyper-parameters of RepOptimizer searched on the two different datasets show similar patterns: the scales of 1×1 and 3×3 branches have similar mean values, which are higher at the beginning of a stage; as the depth increases, the mean values of scales at the identity branch vary insignificantly.

