# OpenReview forum: "Re-parameterizing Your Optimizers rather than Architectures"
_ICLR.cc/2023/Conference — ICLR 2023 poster_

### Official Review · Reviewer_96Hr · 2022-10-24

**Confidence:** 5
**Correctness:** 2
**Technical Novelty And Significance:** 2
**Empirical Novelty And Significance:** 2
**Recommendation:** 3

**Clarity, Quality, Novelty And Reproducibility:**

The definition of the prior knowledge is unclear since it has vague arguments. As the authors mention, the proposed approach requires employment of the prior knowledge via modifying the gradients according to a set of model-specific hyper-parameters. Therefore, it is not clear how to apply the proposed method to other architectures, models, optimizers and tasks.

For instance, I am not sure how to apply the proposed prior knowledge template to WRNs, U-Nets, MobileNets, Transformers, LSTMs/RNNs, GANs etc. for image synthesis/super-resolution, pose estimation, NLP and speech processing/recognition tasks in addition to image classification/detection/segmentation.

Moreover, the proposed approach should be explored together with other optimization methods, like vanilla Adam, RMSProp, etc. and manifold valued optimizers such as Riemannian SGD and Adam, since these optimizers also re-parameterize gradients during optimization.

Theoretical analyses should be further improved. For instance, the proposition in the appendix only tells that there exists a transformation which can provide similar outputs for scaled kernels. However, it does not tell how estimate the optimal transformation. Indeed, a trivial solution would be a transformation rescaled by inverse scaling parameters. However, this solution does not exhibit an improvement.

**Strength And Weaknesses:**

The paper introduces a new approach to reparameterize gradients and parameters according to the prior knowledge “adding up the inputs and outputs of several branches weighted by diverse scales”.

Although this proposal is interesting, it is more like a general idea instead of a particular method. Therefore, the main weakness of the paper is the limitation of generalization of the proposal to different models with different architectures, optimizers and tasks. The proposed method should be also compared with other other reparameterization and optimizer design methods as well.


**Summary Of The Paper:**

This paper proposes incorporating model-specific prior knowledge into optimizers by modifying the gradients according to a set of model-specific hyper-parameters. The proposed methods are employed on VGG architectures using SGD and AdamW optimizers for image classification and segmentation datasets.

**Summary Of The Review:**

The paper introduces a new approach to reparameterize gradients and parameters according to the prior knowledge “adding up the inputs and outputs of several branches weighted by diverse scales”.

However, there are various major and minor issues with the paper. Therefore, the paper is not ready for publication without fixing these issues.

---

> ### Author Response · Authors · 2022-11-16
> **Response to Reviewer 96Hr**
>
> We sincerely thank the reviewer for offering constructive comments. We have revised the paper accordingly. Please see the text marked in blue in the revised paper. In the discussion phase, please enlighten us on anything we can do to improve the score.
>
> **Clarity**
>
> We agree that prior knowledge needs a clear definition. **We have added that to the first page of the revised paper.**
>
> **Generality**
>
> The reviewer raised a concern about the potential generality. As discussed in the last section, the proposed implementation (making the training dynamics of a single operator **equivalent to** a complicated block) relies on the linearity of operations (and it applies to **any linear additive structure**, as shown in Appendix A and the application to the MLP model in Appendix D), but our methodology of shifting the structural priors into the training process may generalize to nonlinear cases, e.g., by using some information derived from the model structure to guide the training.
>
> Though we cannot experiment with every task and every model, we would like to highlight that the method can be used to facilitate the training and quantization of many models on many tasks. Researchers are intensively studying linear structures, providing more use cases for RepOptimizers: YOLOv6 [1], YOLOv7 [2] and PP-YOLOE+ [3] use a RepVGG-like backbone or linear heads. Other representatives include Non-deep Networks [4], MobileOne [5] and RepGhost [6], which heavily use linear structures.
>
> In summary,
>
> 1) by the task, our method can be used for numerous linear structures used in image classification, object detection (Table 7, YOLOv6 [1], YOLOv7 [2] and PP-YOLOE+ [3], RepDarkNet [8] and ReYOLO [9]), semantic segmentation (Table 7, RepUNet [7]), voice recognition [10, 11, 12], image super-resolution [14-19], protein prediction [20], object tracking [21], point cloud [22], defect detection [23], etc.
>
> 2) by the architecture, our method can be combined with linear structures in general dense CNN (RepOpt-VGG), MLP (Appendix D),  lightweight model [6], UNet [7], etc.
>
> The complete reference list is presented in the following comment due to the limitation of characters.
>
> **More comparison with Structural Re-param**
>
> For a concrete example, we design RepOptimzier to simplify RepGhostNet (a lightweight model with Structural Reparam) without degrading the accuracy **(added in Appendix E)**.
>
> **Comparison with other optimizers**
>
> Thanks for the suggestion, but we already discussed the other model-agnostic optimizers in the initial paper.
>
> 1) For training RepOpt-VGG, we compared RepOptimizer and the other model-agnostic optimizers, including momentum SGD, AdamW and Adafactor (Table 4).
>
> 2) The methodology of Gradient Re-parameterization can be combined with multiple update rules: the RepOptimizer for RepOpt-VGG uses SGD as the update rule, while the RepOptimizer for training the MLP model uses the AdamW update rule (Section 4.2 and Appendix D).
>
> We appreciate the reviewer's suggestions and agree more comparisons are helpful, so **we have added the results with vanilla Adam and RMSprop in Table 6**. It turns out that the models trained with multiple model-agnostic optimizers are inferior to the RepOptimizer-trained one, suggesting the update rule is not the key, but the model-specific prior knowledge is. We cannot use Riemannian SGD because it is not designed for deep CNN, and we have not seen it used on practical deep models.
>
> **Theoretical analysis**
>
> We would like to highlight that the outputs are **not similar** but mathematically identical. The transformation is **equivalent**: the simple model trained with a RepOptimizer will yield **identical** results to a complicated model trained with a regular optimizer **at any training iteration**, which we have highlighted in Section 1, Section 3, Section 4 and Appendix A.
>
> We are not sure what **"optimal transformation"** means and what we are expected to estimate since we did not claim anything optimal nor estimate anything. If the meaning is that the searched scales cannot be proven to be optimal, we would agree because that is the major weakness of deep learning - it is hardly possible to prove some parameters in a big deep model optimal or estimate the gap. We believe only a complete mathematical understanding of deep learning can possibly solve this problem, which may take years of effort from numerous researchers. If that is not the actual meaning, we wish the reviewer could further enlighten us in the discussion phase.
>
> Besides, the reviewer commented that **"a trivial solution would be a transformation rescaled by inverse scaling parameters."** We wish to be enlightened on what "inverse scaling parameters" mean to understand why it is trivial and address this concern. We also wish the reviewer could enlighten us on the meaning of **"this solution does not exhibit an improvement"** because this sentence seems to comment on some experimental result, but we are not sure what it refers to.

---

> ### Author Response · Authors · 2022-11-16
> **(full reference list)**
>
> This comment contains the reference list due to the limitation of characters per comment.
>
> Representative architectures:
>
> [1] Li, C., Li, L., Jiang, H., Weng, K., Geng, Y., Li, L., ... & Wei, X. (2022). YOLOv6: a single-stage object detection framework for industrial applications. arXiv preprint arXiv:2209.02976.
>
> [2] Wang, C. Y., Bochkovskiy, A., & Liao, H. Y. M. (2022). YOLOv7: Trainable bag-of-freebies sets new state-of-the-art for real-time object detectors. arXiv preprint arXiv:2207.02696.
>
> [3] Xu, S., Wang, X., Lv, W., Chang, Q., Cui, C., Deng, K., ... & Lai, B. (2022). PP-YOLOE: An evolved version of YOLO. arXiv preprint arXiv:2203.16250.
>
> [4] Goyal, A., Bochkovskiy, A., Deng, J., & Koltun, V. (2021). Non-deep networks. arXiv preprint arXiv:2110.07641.
>
> [5] Vasu, P. K. A., Gabriel, J., Zhu, J., Tuzel, O., & Ranjan, A. (2022). An Improved One millisecond Mobile Backbone. arXiv preprint arXiv:2206.04040.
>
> [6] Chen, C., Guo, Z., Zeng, H., Xiong, P., Dong, J. (2022). RepGhost: A Hardware-Efficient Ghost Module via Re-parameterization. arXiv preprint arXiv:2211.06088.
>
> Applications:
>
> Segmentation (UNet):
>
> [7] Zheng, Y., Er, M. J., Yi, G., & Shen, S. (2021, July). RepUNet: A Fast Image Semantic Segmentation Model Based on Convolutional Reparameterization of Ship Satellite Images. In 2021 6th International Conference on Automation, Control and Robotics Engineering (CACRE) (pp. 461-465). IEEE.
>
> Other detection models:
>
> [8] Zhou, L., Zheng, C., Yan, H., Zuo, X., Liu, Y., Qiao, B., & Yang, Y. (2022). RepDarkNet: A Multi-Branched Detector for Small-Target Detection in Remote Sensing Images. ISPRS International Journal of Geo-Information, 11(3), 158.
>
> [9] Zhang, J., Zheng, Z., Xie, X., Gui, Y., & Kim, G. J. (2022). ReYOLO: A traffic sign detector based on network reparameterization and features adaptive weighting. Journal of Ambient Intelligence and Smart Environments, (Preprint), 1-18.
>
> Voice recognition:
>
> [10] Ma, Y., Zhao, M., Ding, Y., Zheng, Y., Liu, M., & Xu, M. (2021). Rep Works in Speaker Verification. arXiv preprint arXiv:2110.09720.
>
> [11] Zhao, Z., Li, Z., Wang, W., & Zhang, P. (2022). The HCCL system for VoxCeleb Speaker Recognition Challenge 2022.
>
> [12] Brown, A., Huh, J., Chung, J. S., Nagrani, A., & Zisserman, A. (2022). VoxSRC 2021: The third VoxCeleb speaker recognition challenge. arXiv preprint arXiv:2201.04583.
>
> Video denoising:
>
> [13] Sharan, S. P., Krishna, A. K., Rao, A. S., & Gopi, V. P. (2022, May). RepAr-Net: Re-Parameterized Encoders and Attentive Feature Arsenals for Fast Video Denoising. In 2022 International Conference on Robotics and Automation (ICRA) (pp. 633-639). IEEE.
>
> Super-resolution:
>
> [14] Li, Y., Zhang, K., Timofte, R., Van Gool, L., Kong, F., Li, M., ... & Fang, J. (2022). Ntire 2022 challenge on efficient super-resolution: Methods and results. In Proceedings of the IEEE/CVF Conference on Computer Vision and Pattern Recognition (pp. 1062-1102).
>
> [15] Zhang, S., Chen, X., & Huang, X. (2022). Lightweight Image Super-Resolution Based on Re-Parameterization and Self-Calibrated Convolution. Computational Intelligence and Neuroscience, 2022.
>
> [16] Luo, J., Si, W., & Deng, Z. (2022). Few-Shot Learning for Radar Signal Recognition Based on Tensor Imprint and Re-Parameterization Multi-Channel Multi-Branch Model. IEEE Signal Processing Letters, 29, 1327-1331.
>
> [17] Wang, L., Li, D., Tian, L., & Shan, Y. (2022). Efficient Image Super-Resolution With Collapsible Linear Blocks. In Proceedings of the IEEE/CVF Conference on Computer Vision and Pattern Recognition (pp. 817-823).
>
> [18] Bhardwaj, K., Milosavljevic, M., O'Neil, L., Gope, D., Matas, R., Chalfin, A., ... & Loh, D. (2022). Collapsible linear blocks for super-efficient super resolution. Proceedings of Machine Learning and Systems, 4, 529-547.
>
> [19] Wang, X., Dong, C., & Shan, Y. (2022). RepSR: Training Efficient VGG-style Super-Resolution Networks with Structural Re-Parameterization and Batch Normalization. arXiv preprint arXiv:2205.05671.
>
> Protein:
>
> [20] Guo, L., He, J., Lin, P., Huang, S. Y., & Wang, J. (2022). TRScore: a 3D RepVGG-based scoring method for ranking protein docking models. Bioinformatics, 38(9), 2444-2451.
>
> Video object tracking:
>
> [21] Yang, F., Zhang, X., & Liu, B. (2022). Video object tracking based on YOLOv7 and DeepSORT. arXiv preprint arXiv:2207.12202.
>
> 3D point cloud:
>
> [22] Tang, K., Chen, Y., Peng, W., Zhang, Y., Fang, M., Wang, Z., & Song, P. (2022). RepPVConv: attentively fusing reparameterized voxel features for efficient 3D point cloud perception. The Visual Computer, 1-12.
>
> Defect detection:
>
> [23] Liu, J., Yao, Z., Guo, S., Xie, H., & Yang, G. (2022). PriRepVGG: Privacy-Preserving 3-Party Inference Framework for Image-Based Defect Detection. Applied Sciences, 12(19), 10168.

---

### Official Review · Reviewer_yhDx · 2022-10-25

**Confidence:** 3
**Correctness:** 3
**Technical Novelty And Significance:** 3
**Empirical Novelty And Significance:** 3
**Recommendation:** 8

**Clarity, Quality, Novelty And Reproducibility:**

Clarity/quality:
The paper is well-written.

Novelty:
see above

Reproducibility:
Authors presented details of experiments and mentioned that they plan to release code. Therefore, it seems that reproducing results would not be difficult.

**Strength And Weaknesses:**

Strength:

The approach of this paper is quite novel. Authors pointed out that existing works mainly focus on structural re-parameterization (SR). Compared to SR, the clear advantage of the proposed RepOptimizer is that the model architecture stays the same during train and test. It makes training faster and lighter without decreasing accuracy (Table 2 and Table 3). The efficacy of Hyper-Search is validated in Table 4 and Table 5. It is impressive to me that the found constants from different search datasets using Hyper-Search ensures similar accuracies on target datasets. The paper showcased two priors, adding up branches (RepOpt-VGG) and locality (RepOpt-MLP), which can be handled by the proposed algorithm on different tasks such as image classification, detection, and segmentation (Table 7). In addition, the parameter distribution in Figure 3 shows that the RepOptimizer is robust to parameter quantization error. Overall, the paper is well-written and presents valuable insights on optimizers with supporting experiments.

Weakness:

As authors mentioned in the conclusion section, the proposed method relies on the linearity of operations. Hence, the prior should be able to be represented using linear operations in CSLA structure. It might limit the applicable domains of this approach. It would be great if authors show that the non-linear operations can be approximated using RepOptimizer without sacrificing accuracy too much. The Hyper-Search requires another training on an additional dataset. It might not be concretely fair to make comparisons to other approaches that use a single target dataset. I am also curious if the proposed method is scalable to large-scale distributed training scenarios.

**Summary Of The Paper:**

This paper describes a way to impose model-specific prior knowledge into optimizers for training neural networks. Given CSLA structure inspired by the prior, authors seek for parameters of Grad Mult tensor using Hyper-Search. The Grad Mult tensor changes back-propagation gradients directly. It allows us to use the same model architecture for training and testing.

**Summary Of The Review:**

See above. Overall the paper is novel, well-written and claims supported.

---

> ### Author Response · Authors · 2022-11-16
> **Response to Reviewer ydDx**
>
> We sincerely thank the reviewer for appreciating this paper and offering constructive comments.
>
> As discussed in the last section, we acknowledge that the present method applies to linear structures, but we would like to highlight that researchers are intensively studying linear structures, providing more use cases for RepOptimizers. For example, YOLOv6 [1], YOLOv7 [2] and PP-YOLOE+ [3] use a RepVGG-like backbone or linear detectors with linear heads. Other representative works include Non-deep networks [4], MobileOne [5] and RepGhost [6], which heavily use linear structures. We would like to further explore our method's application to more structures in the future.
>
> As for the potential unfair comparison brought by Hyper-Search, we would like to note that the target model only inherits the constants from the Hyper-Search but not the majority of trained parameters (e.g., convolutional kernels) for the fair comparison. In practice, initializing the target model with all the trained parameters makes no improvements since the Hyper-Search dataset is too small compared to the target dataset. Moreover, considering the additional computational cost is only about 0.18% of training, as analyzed in Section 4.1, and the training speed of our method improves significantly (1.8x), our method still holds excellent advantages both in memory consumption and training speed.
>
> Due to the limitation of computational resources, we did not conduct large-scale distributed training experiments. As the mathematical basis (CSLA=GR, Appendix A) does not depend on concrete training scenarios, our method is expected to work with large-scale distributed training.
>
> [1] Li, C., Li, L., Jiang, H., Weng, K., Geng, Y., Li, L., ... & Wei, X. (2022). YOLOv6: a single-stage object detection framework for industrial applications. arXiv preprint arXiv:2209.02976.
>
> [2] Wang, C. Y., Bochkovskiy, A., & Liao, H. Y. M. (2022). YOLOv7: Trainable bag-of-freebies sets new state-of-the-art for real-time object detectors. arXiv preprint arXiv:2207.02696.
>
> [3] Xu, S., Wang, X., Lv, W., Chang, Q., Cui, C., Deng, K., ... & Lai, B. (2022). PP-YOLOE: An evolved version of YOLO. arXiv preprint arXiv:2203.16250.
>
> [4] Goyal, A., Bochkovskiy, A., Deng, J., & Koltun, V. (2021). Non-deep networks. arXiv preprint arXiv:2110.07641.
>
> [5] Vasu, P. K. A., Gabriel, J., Zhu, J., Tuzel, O., & Ranjan, A. (2022). An Improved One millisecond Mobile Backbone. arXiv preprint arXiv:2206.04040.
>
> [6] Chen, C., Guo, Z., Zeng, H., Xiong, P., Dong, J. (2022). RepGhost: A Hardware-Efficient Ghost Module via Re-parameterization. arXiv preprint arXiv:2211.06088.

---

### Official Review · Reviewer_FagT · 2022-10-29

**Confidence:** 4
**Correctness:** 3
**Technical Novelty And Significance:** 3
**Empirical Novelty And Significance:** 3
**Recommendation:** 8

**Clarity, Quality, Novelty And Reproducibility:**

They intend to make the code available. The results are novel and the paper is clear and understandable.

**Strength And Weaknesses:**

Strength:
- Proposes a novel method for gradient reparameterization which allows incorporation of certain model priors equivalent to changes in the training architecture more efficiently.
- Achieves strongs results on improving training speed while achieving results comparable to prior work while also being easier to quantize.
- Demonstrated that the hyperparameters seem to be dataset agnostic even with different input sizes.


Weakness:
- The method requires relatively expensive hyperparameter tuning on a per-model basis.
- Does not conduct significant analysis on the actual values of these important hyperparameters.


Comments:
The paper may benefit from more runtime analysis of the hyperparameter search method in terms of real time measurements as the major empirical improvement of this method is on training speed.

The paper may benefit from stronger analysis of the search process for hyperparameters. While the reinitialization is justified with the connection to NAS and it could be understandable how it alters the training dynamics, it would benefit from further analysis. Is my understanding correct that you conduct the hyper-parameter search similar to the first order approximation of DARTS no the second order search method? It would be interesting to how the model performs if you reinitialize the network weight, but still allow those parameters to be tuned. Mainly it would greatly improve the understanding of this hyperparameter which is integral to the method if you analyzed how the values and distribution of those hyperparameters changed during the search process as well as how they differ for different layers at different depths and channel sizes.


**Summary Of The Paper:**

The paper proposes a method they call Gradient Reparamerization which allows them to efficiently add priors to the model without additional training structures other than adding a Gradient Multiplier generated based on tuned hyperparameters. They are able to train models that perform on par or better than recent models with a network with a simple structure, and high training and inference efficiency. They achieve comparable results to RepVGG with significantly faster and more memory efficient training with the same inference model. They demonstrate that their models are naturally much easier to quantize compared to structurally re-parameterized models. They derive their method by proposing that based on prior work performance improvements can be made with parallel convolutions multiplied by a single channelwise scaler which are then summed. They derive that the training dynamics are exactly equivalent to by channelwise rescaling the gradients for a single convolution and propose Gradient Reparamerization which involves this rescaling as well as a scale-aware initialization. This method does require hyperparameter tuning of the multipliers, but they find that they are dataset agnostic even to different sized inputs and need only be specialized for each model architecture. They tune the hyperparameter setting the channelwise scalers to be trainable instead of fixed and optimizing it along with the weight parameters. They demonstrate that they can achieve good results on Imagenet by first hyperparameter tuning on cifar10 which theoretically should only require approximately 0.18% of training on full Imagenet.

**Summary Of The Review:**

Overall this paper makes a strong contribution. Their novel method to shift architecture priors into the optimizer opens up a new direction to consider for researchers. They demonstrate strong empirical results in improving training efficiency. The main downside is that to achieve best results, their method requires a hyperparameter tuning step and the work would benefit significantly from more analysis of this area. The presentation is clear and they intend to make the code publicly available.

---

> ### Author Response · Authors · 2022-11-16
> **Response to Reviewer FagT**
>
> We sincerely thank the reviewer for appreciating this paper and offering constructive comments. We have revised the paper accordingly.
>
> Weakness 1 (cost of Hyper-Search)
>
> We would like to clarify that the cost of Hyper-Search is negligible. As shown in Section 4.1, the Hyper-Search only demands a small dataset like CIFAR-100, which has only 50k images and an input resolution of 32, while ImageNet has 1281k images and a resolution of 224. We Hyper-Search on CIFAR-100 for 240 epochs and train on ImageNet for 120 epochs. Considering the Hyper-Search model has extra 1x1 convolutions, the cost of Hyper-Search, compared to the ImageNet training, is roughly only (50/1281)*(240/120)*(32/224)^2*((3x3+1x1)/(3x3)) = 0.18%. Therefore, the hyperparameter search does not hinder the practical application of our method. We did not report the time cost of the Hyper-Search because we train on ImageNet with power GPUs (2080Ti) while searching on CIFAR100 cannot fully utilize the computational power (as the input resolution is too small), so the comparison would be unfair. In practice, we may Hyper-Search with lower-end and cheaper devices and train with expensive but powerful ones.
>
> Weakness 2 (runtime analysis of the hyperparameter search)
>
> Thanks for the suggestion! We agree that such runtime analysis will strengthen this paper. **Please see the added results in Appendix F of the revised paper (marked in blue)**. Precisely, we depict the mean and standard deviation of the scales searched on different datasets (CIFAR-100 and Caltech256) in Fig. 6 in Appendix F. As the Hyper-Search processes, the values of scales searched on different datasets show similar trends and ranges, demonstrating that the hyperparameters are dataset-agnostic.
>
> Question 1 (How does the model perform if we still allow those parameters to be tuned?)
>
> The accuracy will match that of RepVGG and RepOpt-VGG, but with no good because
>
> 1) The model will not be friendly to quantize, as we will still need to convert the trained parameters into a single convolution, and conversion results in poor quantization performance (as discussed in Section 4.4 and Appendix C)
>
> 2) The training will be slower than RepOpt-VGG, as tuning such extra parameters will slow down the forward and backward propagation.
>
> Question 2 (Is Hyper-Search similar to the first-order approximation of DARTS?)
>
> To be more specific, we compare our method to DARTS only because we use the trained values of the scales as the expected values of scales, which is similar to the high-level idea of DARTS. DARTS also obtains the architectural parameters via a training process, where the parameters and hyper-parameters are alternatively updated. But our method significantly differs from DARTS as
>
> 1) Most importantly, Hyper-Search is not NAS because we only desire the trained values of some parameters, not architectural hyperparameters. So Hyper-Search is not similar to the first-order approximation of DARTS - it is just the simplest training process. We just use an analogy for introducing the high-level idea (the trained values of some parameters are the values the model expects them to be), as discussed in Section 3.3. **We have added the explanations to Section 3.3 of the revised paper.**
>
> 2) In Hyper-Search, all the parameters are simply trained end-to-end on the same search dataset (while DARTS updates the hyper-parameters with a validation set and the parameters with the training set).

---

> > ### Comment · Reviewer_FagT · 2022-11-21
> > **Response to Authors**
> >
> > I would like to thank the authors for their response to my questions. I believe this is a strong paper and will keep my accept rating.

---

### Official Review · Reviewer_Hu4S · 2022-10-31

**Confidence:** 3
**Correctness:** 3
**Technical Novelty And Significance:** 3
**Empirical Novelty And Significance:** 3
**Recommendation:** 6

**Clarity, Quality, Novelty And Reproducibility:**

The paper is well structured but a bit difficult to read due to either long paragraphs or too much detail. It is not clear whether the approach can be easily reproduced given the complexity of the approach and hyper-search. The originality of the work is good.

**Strength And Weaknesses:**

Strength
- the idea of integrating prior knowledge of architectural design into the design of optimizer is interesting and worth exploring.
- the authors provide a concrete example of implementing the idea with a plain CNN network, specifically demonstrating the effectiveness of RepOptimizers by showing RepOpt-VGG closely matches the accuracy of RepVGG.

Weakness
- the model prior is not well defined.
- the advantage of the proposed method in comparison with traditional architecture design is not quite clear.
- It is not clear whether all architecture design can be transferred to equivalent optimization design. The authors only provide examples such as converting residual connections, but it is not clear for other architectures whether the RepOptimizers can perform the same. If not, it makes the application of the proposed approach quite limited.
- The hyper-parameters searched on CIFAR-100 are transferable to ImageNet may not be enough to support the claim that “the RepOptimizers may be model-specific but dataset-agnostic”.


**Summary Of The Paper:**

This paper proposes to incorporate model-specific prior knowledge into optimizers by modifying the gradients according to a set of model-specific hyper-parameters. They name the method as Gradient Re-parameterization, and the optimizers are named RepOptimizers. They show that a VGG-style plain network can be trained with the proposed optimizer to perform on par with or better than the recent well-designed models.


**Summary Of The Review:**

The authors explore the space of optimizer design that is equivalent to architecture prior. This paper performs initial exploration and demonstration replacing the residual connections, which could inspire future works in this direction.

---

> ### Author Response · Authors · 2022-11-16
> **Response to Reviewer Hu4S**
>
> We sincerely thank the reviewer for the constructive comments. We have revised the paper accordingly (please see the revised version). Please enlighten us on anything we can do to further improve the score in the discussion phase.
>
> Weakness 1
>
> From a general machine-learning perspective, prior knowledge refers to all information about the problem and the training data [1]. This concept is closely related to inductive bias (since a reasonable inductive bias should depend on prior knowledge). The inductive bias of a learning algorithm is the set of assumptions that the learner uses to predict outputs of given inputs that it has not encountered [2], i.e.,  necessary assumptions about the nature of the target function. Specific to the structural design of models, since we have not encountered any data sample while designing the model, any structural designs can be regarded as inductive biases, which reflect our prior knowledge about the data. In other words, prior knowledge can be described as "though I have not trained the model on the data, I believe that if I use such structures, the model will probably perform better." Such explanations have been **added to the first page**.
>
> For example, in the case of RepOpt-VGG, the prior (performance may be improved by adding up the inputs and outputs of several branches weighted by diverse scales) is supported by the success of numerous models, including ResNet and every other model with shortcuts, Inception and ResNeSt [3] (adding up outputs from branches with different kernel sizes), RepVGG (adding up linear branches), etc.
>
>
> Weakness 2
>
> We show the advantages of our method over traditional structural design from three aspects: 1) Compared with RepVGG, RepOpt-VGG consumes less memory and trains faster (1.8x with the respective MaxBS) with comparable accuracy, showing a clearly better trade-off between training efficiency and accuracy; 2) For the comparison with EfficientNets, RepOpt-VGG outperforms both in accuracy and inference speed, even though EfficientNets consumed much more training resources. 3) Models trained with RepOptimizers are much easier to quantize than the structural reparameterization approach (75.89 v.s. 54.55 quantized accuracy).
>
> Weakness 3
>
> As discussed in the last section, the proposed concrete implementation (making the training dynamics of a single operator equivalent to a complicated block) relies on the linearity of operations (and it applies to any linear additive structure, as shown by Proposition 1 and the application to the MLP model in Appendix D), but our methodology of shifting the structural priors into the training process may generalize to nonlinear cases, e.g., by using some information derived from the model structure to guide the training.
>
> Moreover, researchers are intensively studying linear structures, providing more use cases for RepOptimizers. For example, YOLOv6 [4], YOLOv7 [5] and PP-YOLOE+ [6] use RepVGG-like backbones or detectors with linear heads. Other representatives include Non-deep Network, RepUNet, RepGhostNet, etc. **We have added an example of using RepOptimizer for RepGhostNet in Appendix E**.
>
> Weakness 4
>
> The fact that the CIFAR-searched hyperparameters transfer to ImageNet is not the only evidence. Table 5 shows the hyperparameters searched on three different search datasets (CIFAR-100, ImageNet and Caltech256) perform similarly on two target datasets (Caltech256 and ImageNet), indicating that different input resolutions and dataset sizes lead to similar properties of hyperparameters. Moreover, the hyperparameters searched on classification datasets can directly transfer to other tasks (Table 7). The consistent and robust performance across multiple datasets demonstrates our methodology to be dataset-agnostic.
>
> Reproducibility
>
> We will make the code, models and reproducible training scripts publicly available.
>
> [1] Krupka, E., & Tishby, N. (2007, March). Incorporating prior knowledge on features into learning. In Artificial Intelligence and Statistics (pp. 227-234). PMLR.
>
> [2] Mitchell, T. M. (1980). The need for biases in learning generalizations (pp. 184-191). New Jersey: Department of Computer Science, Laboratory for Computer Science Research, Rutgers Univ.
>
> [3] Zhang, H., Wu, C., Zhang, Z., Zhu, Y., Lin, H., Zhang, Z., ... & Smola, A. (2022). Resnest: Split-attention networks. In Proceedings of the IEEE/CVF Conference on Computer Vision and Pattern Recognition (pp. 2736-2746).
>
> [4] Li, C., Li, L., Jiang, H., Weng, K., Geng, Y., Li, L., ... & Wei, X. (2022). YOLOv6: a single-stage object detection framework for industrial applications. arXiv preprint arXiv:2209.02976.
>
> [5] Wang, C. Y., Bochkovskiy, A., & Liao, H. Y. M. (2022). YOLOv7: Trainable bag-of-freebies sets new state-of-the-art for real-time object detectors. arXiv preprint arXiv:2207.02696.
>
> [6] Xu, S., Wang, X., Lv, W., Chang, Q., Cui, C., Deng, K., ... & Lai, B. (2022). PP-YOLOE: An evolved version of YOLO. arXiv preprint arXiv:2203.16250.

---

### Author Response · Authors · 2022-11-17
**Response to all the reviewers**

Dear Reviewers,

We would like to thank all the reviewers for their feedback and help in improving our work. We are glad that they appreciate the novelty (Reviewer yhDx), originality (Reviewer Hu4S), strong results (Reviewer FagT), good writing and presentation (Reviewer yhDx), etc.

**We have revised the paper as suggested (marked in blue in the revised paper)**, including

1) A clear definition of prior knowledge

2) Results of generalizing RepOptimizer to RepGhostNet (Appendix E), as another evidence supporting the generality of our method

3) More discussions and results with model-agnostic optimizers (Section 4.2, Table 4)

4) Analysis of the scale values in the Hyper-Search process (Appendix F)

**We have responded to every reviewer separately. In the discussion phase, please enlighten us on anything we can do to further improve the paper.**

Best Regards,

Authors

---

### Decision · Program_Chairs · 2023-01-20

**Decision:**

Accept: poster

**Justification For Why Not Higher Score:**

The overall scope might not be sufficient to justify a spotlight.  The positive reviewers who gave a score of 8 still point to some potential drawbacks of the method; the AC is confident in the direction, but not magnitude of those scores.

**Justification For Why Not Lower Score:**

This appears to be a novel and interesting idea with experimental backing.

**Metareview: Summary, Strengths And Weaknesses:**

The paper proposes a technique that re-parameterizes optimizers while keeping a simpler model architecture, as an alternative to a more complicated model architecture with a standard optimizer.  Three out of four reviewers are positive, noting the idea is novel and interesting, and experiments demonstrate practical benefits.  The Area Chair agrees with this plurality of reviewers.

**Note From Pc:**

if the above contains the word "oral" or "spotlight" please see: "oral" presentation means -> notable-top-5% and "spotlight" means -> notable-top-25%. As stated in our emails, we are disassociating presentation type from AC recommendations